# Regret-Based Federated Causal Discovery with Unknown Interventions

**Federico Baldo** [1]   **Charles K. Assaad** [1]

## Abstract

Most causal discovery methods recover a completed partially directed acyclic graph (CPDAG) representing a Markov equivalence class from observational data. Recent work has extended these methods to federated settings to address data decentralization and privacy constraints, but often under idealized assumptions that all clients share the same causal model. Such assumptions are unrealistic in practice, as client-specific policies, for instance, across hospitals, naturally induce heterogeneous and unknown interventions. In this work, we address federated causal discovery under unknown client-level interventions. We propose I-PERI, a novel federated algorithm that first recovers the CPDAG common to all clients and then orients additional edges by exploiting structural differences induced by interventions across clients. This yields a tighter equivalence class, which we call the $\Phi$-Markov Equivalence Class, represented by an augmented version of the CPDAG, namely, a $\Phi$-CPDAG. We provide theoretical guarantees on the convergence of I-PERI, as well as on its privacy-preserving properties, and present empirical evaluations demonstrating the effectiveness of the proposed algorithm.

## 1. Introduction

Discovering causal structures from observational and interventional data is a challenging task at the core of many scientific disciplines. Indeed, recovering causal relationships among variables is crucial for estimating causal effects, as it enables the identification of confounding variables (Pearl, 2009; Spirtes et al., 2001; Miguel et al., 2023). Many causal discovery algorithms (Spirtes et al., 2001; Colombo & Maathuis, 2014; Runge, 2020; Assaad et al., 2022) aim to recover, from observational data, a completed partially

directed acyclic graph (CPDAG) (Chickering, 2003), which represents the Markov equivalence class (MEC) (Verma & Pearl, 1990) of an underlying causal graph. These methods typically assume access to a centralized dataset (Mooij et al., 2020; Li et al., 2023). In many real-world scenarios, however, data are inherently decentralized and cannot be pooled together due to practical or legal constraints. This has motivated growing interest in federated learning (Kairouz et al., 2021), where data are distributed across multiple clients, for example, hospitals, institutions, or organizations, each of which holds its own local dataset (Zanga et al., 2025).

In Federated Causal Discovery (FCD), a central server coordinates the discovery of causal graphs by aggregating information communicated by the clients, without accessing their individual datasets. Typically, these methods extend traditional causal discovery approaches to distributed settings, inheriting the underlying methodological paradigms and assumptions. The existing methods fall into three categories: constraint-based (Li et al., 2024; Wang et al., 2025; Huang et al., 2023; Guo et al., 2024a;b; Wang et al., 2023), score-based (Mian et al., 2023; Ye et al., 2024), and optimization-based (Abyaneh et al., 2024; Ng & Zhang, 2022; Gao et al., 2023; Yang et al., 2024; Liu et al., 2024). These works addressed the underlying challenges of FCD in an idealized setting, where all clients' data are assumed to be generated by the same causal model and to be unaffected by interventions, thereby sharing an identical causal graph. This assumption is frequently unrealistic in real-world applications. For instance, in healthcare scenarios, different hospitals may adopt distinct treatment policies, diagnostic protocols, or patient inclusion criteria. These differences naturally induce client-specific interventions and lead to heterogeneous causal mechanisms. Abyaneh et al. (2024) addresses this problem integrating *known* interventions in the discovery process. However, to the best of our knowledge, no existing work has addressed this problem in presence of *general unknown interventions* at the client-level. Additionally, beyond decentralization, federated approaches are particularly appealing in domains where data sharing is restricted by regulatory requirements, data ownership constraints, or privacy concerns. However, the challenge of designing inherently privacy-preserving FCD algorithms has been largely overlooked in many existing works.

In this paper, we propose I-PERI, a novel FCD algorithm ca-

---

[1]Sorbonne Université, INSERM, Institut Pierre Louis d'Epidémiologie et de Santé Publique, F75012, Paris, France. Correspondence to: Federico Baldo <federico.baldo@inserm.fr>.

*Proceedings of the 43rd International Conference on Machine Learning*, Seoul, South Korea. PMLR 306, 2026. Copyright 2026 by the author(s).

pable of handling unknown interventions at the client-level while ensuring differential privacy. I-PERI is an extension of the recently proposed PERI algorithm (Mian et al., 2023); this method assumes that all clients share the same causal structure, and it is capable of discovering the CPDAG common to all clients from purely observational data in a differentially private fashion. I-PERI leverages interventional data by introducing a two-phase approach: first, it learns the CPDAG corresponding to the graph from which all client graphs are derived, for instance, through interventions; then, it refines the learned structure by orienting additional edges, exploiting structural differences induced by interventions across clients.

We want to emphasize that this work differs from existing approaches to causal discovery with interventional data. In Hauser & Bühlmann (2012); Yang et al. (2018) it is assumed that interventions are known, or can be derived. This assumption substantially simplifies the discovery problem, allowing us to identify a tighter class, called $\mathcal{I}$-MEC. However, it is often unrealistic in practical federated settings. In contrast, we consider a setting in which interventions are unknown to the server and may differ across clients. More recently, Jaber et al. (2020); Li et al. (2023); Squires et al. (2020); Wang et al. (2022) have established important results for causal discovery from datasets with unknown interventions. However, they assume that all datasets can be pooled centrally, allowing direct comparison across intervention regimes. This enables stronger identifiability results than those attainable in a federated environment. Moreover, they focus on interventions that do not modify the structure of the causal graph, i.e., parametric interventions. In this paper, we identify an equivalence class stricter than the standard MEC, namely, the $\Phi$-MEC, which is the best one we can recover in a federated setting with unknown interventions at the client-level while preserving differential privacy[1]. Our contributions are threefold:

- We introduce the $\Phi$-Markov equivalence class under a set of unknown interventions in a federated setting, which forms a tighter equivalence class than the standard MEC, along with a representative graph for this class, the $\Phi$-CPDAG.

- We propose I-PERI, a novel algorithm for FCD that effectively combines observational and interventional data from multiple clients, ensuring differential privacy, without requiring knowledge of the interventional targets.

- We provide theoretical guarantees on the convergence of I-PERI, as well as on its privacy-preserving properties.

---

[1] We restrict to scenarios in which the client graph is not shared and only regrets are exchanged.

The remainder of this paper is organized as follows: Section 2 introduces background and preliminaries; Section 3 presents I-PERI and provides theoretical details on the $\Phi$-MEC; Section 4 includes the experimental evaluation of the method; finally, Section 5 concludes the paper and discusses future directions. Proofs of theoretical results and additional experimental details are provided in the Appendix.

## 2. Preliminaries & Problem Setup

**Causal Graphs.** A causal directed acyclic graph (DAG) is a graphical representation of causal relationships between random variables. More formally, $G = (\mathbb{V}, \mathbb{E})$ is a DAG where $\mathbb{V} = V_1, V_2, \ldots, V_d$ is the set of nodes representing random variables and $\mathbb{E} \subseteq \mathbb{V} \times \mathbb{V}$ is the set of directed edges representing direct causal relationships between the variables. We denote with $P_\emptyset$ the joint probability distribution over $\mathbb{V}$ that factorizes according to the causal DAG $G$ as follows:

$$P_\emptyset(\mathbb{V}) = \prod_i P_\emptyset(V_i | V_{Pa_i^G}),$$

where $Pa_i^G$ are the parents of $V_i$ in $G$. A key notion in DAGs is d-separation, denoted $\perp\!\!\!\perp_G$, which encodes the conditional independencies implied by the graph. Additionally, we define a v-structure, or unshielded collider, as a specific configuration of three nodes $(V_i, V_j, V_k)$ such that $V_i \to V_j \leftarrow V_k$ and there is no edge between $V_i$ and $V_k$. If a collider has an edge between $V_i$ and $V_k$, it is referred to as a shielded collider. *Causal discovery* aims to learn the causal DAG $G$ or rather a partially oriented version of the DAG from data sampled from the joint distribution $P_\emptyset$. In general, a causal DAG or its partially oriented version cannot be discovered from observational data without additional assumptions. In this work, we adopt two standard assumptions commonly used in discovery methods. The first is the causal sufficiency assumption, which posits that all common causes of the observed variables are themselves observed, i.e., there are no latent confounders. The second is the faithfulness assumption, which states that all and only the conditional independencies present in the distribution $P_\emptyset$ are implied by d-separation.

FCD aims to learn a global graph in a central server, given a set of distributed clients holding the data. In this paper, we focus on federated algorithms that provide privacy-preserving guarantees by design, without relying on cryptographic techniques. For example, let us consider multiple hospitals conducting different clinical trials; in this context, we might want to learn a graph from the joint data while preserving patient privacy. We assume to have $K$ clients, each holding a local dataset $\mathbb{D}^k$, for $k \in 1, \ldots, K$. Each dataset involve the same set of variables $\mathbb{V} = V_1, \ldots, V_d$, and may contain different numbers of samples, $n^1, \ldots, n^K$. In most existing works, clients' data are assumed to be generated

from the same underlying causal DAG $G$ over $\mathbb{V}$, which is ideally the one we want to recover at the server-level. This assumption is often unrealistic in practice. In our example, different clinical trials might involve distinct treatment protocols, eligibility criteria, and intervention strategies. As a result, patient data are generated under hospital-specific interventions, leading to heterogeneous causal structures.

**Interventions.** The causal DAG, alongside the joint distribution, allows us to describe interventions on the system. In the scope of this work, we consider *general interventions*, meaning that an intervention can be: either *parametric*, where the functional form of the conditional distribution of the intervened variable is changed, or *structural*, where the causal mechanism of the intervened variable is replaced, removing some or all incoming edges to the intervened node. Note that, unlike most definitions of structural interventions in the literature, which assume that all incoming edges to an intervention target are removed, we adopt a more general notion in which only a subset of the incoming edges may be removed. As shown by Assaad et al. (2023); Shpitser & Tchetgen (2016), this can occur in real-world settings. Note that parametric interventions in this setting do not modify the causal DAG.

**Definition 2.1** (General intervention & Mutilated graph). Under general interventions on a target set $\mathbb{I} \subseteq \mathbb{V}$, such that $\mathbb{I} = \mathbb{P} \cup \mathbb{S}$, where $\mathbb{P}$ is the set of parametric interventions and $\mathbb{S}$ the set of structural interventions, the post-intervention distribution $P_\mathbb{I}$ is given by:

$$P_\mathbb{I}(\mathbb{V}) = \prod_{V_i \in \mathbb{S}} P_\mathbb{S}(V_i | Pa_i^{G_\mathbb{S}}) \prod_{V_j \in \mathbb{P}} P_\mathbb{P}(V_j | Pa_j^G)$$
$$\prod_{V_k \notin \mathbb{I}} P_\emptyset(V_k | Pa_k^G),$$

where $G_\mathbb{S}$ is the mutilated graph of $G$, which is obtained by removing all or part of the incoming edges to the nodes in $\mathbb{S}$.

For the remainder of this paper, we focus primarily on structural interventions, as they represent the most significant and challenging application scenario for our approach. Nevertheless, the algorithm introduced in this paper is sound even in the presence of parametric interventions or no interventions. For more details, see Section 5.

In the scope of this paper, we consider clients that may be subject to *unknown general interventions* at the client-level. We denote a family of unknown intervention targets for a set of clients as $\mathbf{\Phi} = \Phi^1, \ldots, \Phi^K$, where $\Phi^k \in \mathbf{\Phi}$ represents a set of interventions on $\Phi^k \subseteq \mathbb{V}$. This means that each client $k$ may have their own distribution $P_{\Phi^k}$ under interventions $\Phi^k$ and mutilated causal DAG, $G_{\Phi^k}$. Moreover, we assume that at least one client holds purely observational data, as stated in the following assumption.

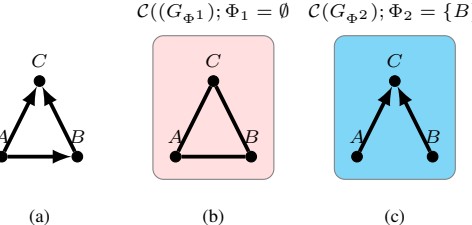

*Figure 1.* (a) True causal DAG; (b) Client CPDAG with no interventions; (c) Client CPDAG of structurally intervened DAG.

**Assumption 2.2.** There exists $k \in 1, \cdots, K$ such that $\Phi_k = \emptyset$.

Note that, unlike other methods assuming known interventions (Hauser & Bühlmann, 2012; Yang et al., 2018), we do not have access to the intervention targets across clients, since it could imply a privacy violation, nor are we inferring the intervention targets (Jaber et al., 2020; Li et al., 2023).

**Greedy Equivalence Search.** Among the different approaches to causal discovery, score-based methods estimate a graph $\hat{G}$ that maximizes a given scoring criterion $L(H, \mathbb{D})$ over a dataset $\mathbb{D}$:

$$\hat{G} = \arg \max_{H \in \mathcal{C}(\mathbb{G})} L(H, \mathbb{D}).$$

where the search is performed over the space of all the possible CPDAGs, $\mathcal{C}(\mathbb{G})$. A popular score-based method is Greedy Equivalence Search (GES) (Chickering, 2003), which performs a greedy search over the space of the MEC of DAGs. GES consists of two phases: a forward phase, where edges are added to the graph to maximize the score, and a backward phase, where edges are removed to further improve the score. The algorithm is guaranteed to return the correct MEC and CPDAG for $n \to \infty$ samples under the assumption of causal sufficiency, faithfulness, and in the presence of a consistent and decomposable scoring criterion (Chickering, 2003):

$$L(H, \mathbb{V}; \mathbb{D}) = \sum_{i=1}^{k} L(V_i, Pa_i^H; \mathbb{D}).$$

The Bayesian Information Criterion (BIC) is an example of a consistent and decomposable scoring criterion (Schwarz, 1978). On a practical level, GES score is computed accounting for both possible directions of an undirected edge; this will be key in the computation of the regret score of our method.

**Regret-based Federated Causal Discovery** Regret-based FCD aims, as formulated in Mian et al. (2022; 2023), to discover causal structure while sharing only regrets, thereby preserving privacy at the client-level. In this framework,

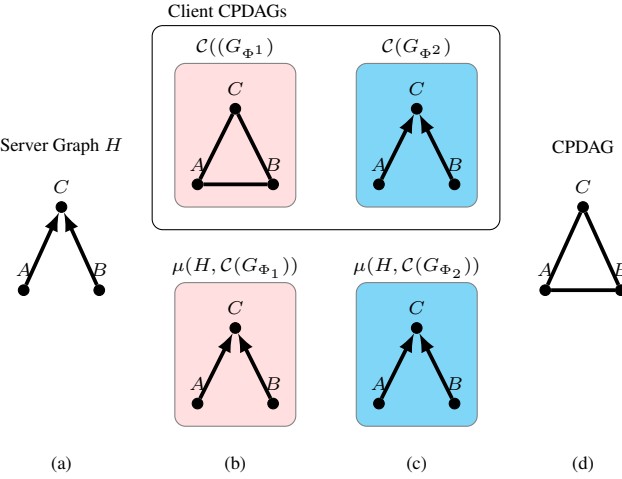

*Figure 2.* (a) Server CPDAG obtained at an intermediate iteration (prior to convergence); (b) client 1 CPDAG (top) and directed masking of the server CPDAG and client 1 CPDAG (bottom); (c) client 2 CPDAG (top) and directed masking of the server CPDAG and client 2 CPDAG (bottom); (d) Target CPDAG result of the first phase of I-PERI.

regrets quantify the discrepancy between a candidate global graph and the client CPDAG. At every moment, the server shares the reconstructed global graph and computes a regret for each client based on a consistent and decomposable scoring function.

**Definition 2.3** (Regret). Let $\mathbb{D}^k$ be the dataset at the client-level, $\mathcal{C}(G)$ the CPDAG of the true DAG, $G$, $L$ a consistent scoring function, and $H$ a candidate global graph. We can define the regret function over $H$ and $\mathcal{C}(G)$ as follows:

$$R_k(H) = L(H, \mathbb{D}^k) - L(\mathcal{C}(G), \mathbb{D}^k). \tag{1}$$

$R_k(H)$ is minimal when $L(H, \mathbb{D}^k)$ and $L(\mathcal{C}(G), \mathbb{D}^k)$ are equal, meaning $H$ and $\mathcal{C}(G)$ are identical.

The algorithm in Mian et al. (2023), called PERI, finds the CPDAG minimizing the worst regret across clients, i.e.:

$$\hat{G} = \arg \min_{H \in \mathcal{C}(\mathbb{G})} \max_k R_k(H). \tag{2}$$

This operation is performed iteratively following the GES algorithm, adding and removing edges to the server graph to minimize the worst regret across clients. The GES algorithm is also applied locally at the client-level to approximate the local CPDAG. In this setting, the assumption is that all data at the client-level are sampled from the same observational distribution, and that the underlying causal DAG is identical across all clients and not subject to interventions. The graph minimizing the worst regret, $\hat{G}$, will be the graph such that $\hat{G} = \mathcal{C}(G)$ for every client $k$ given $n_1, \ldots, n_k \to \infty$, per Theorem 1 and Corollary 3 of Mian et al. (2023).. This paper extends the PERI algorithm in three directions to: (i)

handle interventional data at the client-level with unknown targets; (ii) identify a tighter equivalence class, namely **Φ**-MEC and its representative **Φ**-CPDAG; (iii) allow the use of any causal discovery algorithm at the client-level that returns a CPDAG, e.g., PC (Spirtes et al., 2001).

# 3. Regret-Based Federated Causal Discovery with Unknown Interventions

The intuition behind Intervention-PERI (or I-PERI), is that the intervened graph at the client-level can have missing edges, i.e., edges removed due to structural interventions. Thus, the best we can recover at the client-level is the CPDAG of the mutilated graph, $\mathcal{C}(G_{\Phi^k})$. This hinders the capability of PERI to recover the true CPDAG at the server-level, since the regret function as defined in Equation (1) would not be guaranteed to converge. However, as shown in Figure 1, intervening on the parents of a shielded collider can generate new v-structures, meaning that the client CPDAG may provide additional information about edge orientations. I-PERI operates in two steps: first, it recovers the CPDAG, $\mathcal{C}(G)$, of the causal DAG from which all client graphs are derived, for instance, through interventions; second, it orients edges in $\mathcal{C}(G)$ based on v-structures present at the client-level due to structural interventions, yielding a more refined equivalence class.

## 3.1. Federated CPDAG

The first step of I-PERI identifies a server-level CPDAG that is common across all clients, meaning that each client CPDAG is either identical to the server graph or corresponds to the CPDAG derived from a mutilated version of the same underlying DAG. In this setting, directly applying the regret function as defined in Equation (1) would not converge, since we have no way to account for missing edges due to structural interventions. To address this issue, we compute the regret function over the masked server graph and the client graph —following Definition 3.1 —as illustrated in Figure 2. The intuition is straightforward: we want to penalize only those edges that are absent in the global graph yet present in the client's graph. Conversely, edges missing at the client-level should not be penalized, as their absence may be due to an intervention. To this end, the masking operation removes from the server graph any edges not present in the client graph; additionally, undirected edges in the server graph are oriented according to the directionality of the corresponding client graph edges.

**Definition 3.1** (Directed-consensus masking). Given two CPDAGs, $G^k = (\mathbb{V}, \mathbb{E}^k)$ and $G^l = (\mathbb{V}, \mathbb{E}^l)$, the directed-consensus masking between the two graphs is defined as follows:

$$\mu(G^k, G^l) = (\mathbb{V}, \mathbb{E}^\mu),$$

where $\mathbb{E}^\mu$ is defined as:

- if $V_i \multimap V_j \in \mathbb{E}^k$ and $V_i \multimap V_j \in \mathbb{E}^l$, then $V_i \multimap V_j \in \mathbb{E}^\mu$.

- if $V_i \not\multimap V_j$ in either $\mathbb{E}^k$ or $\mathbb{E}^l$, then $V_i \not\multimap V_j \in \mathbb{E}^\mu$.

- if $V_i \rightarrow V_j \in \mathbb{E}^k$ and $V_i - V_j \in \mathbb{E}^l$, or vice versa, then $V_i \rightarrow V_j \in \mathbb{E}^\mu$.

where $\multimap$ denotes an undirected edge, $-$, or a directed one, $\rightarrow$.

Definition 3.1 subsumes a notion of inclusion, in which a directed edge between two vertices, $V_i \rightarrow V_j$, is always included in an undirected one, $V_i - V_j$.

**Definition 3.2** (Inclusion between CPDAGs). Given two CPDAGs, $G^k = (\mathbb{V}, \mathbb{E}^k)$ and $G^l = (\mathbb{V}, \mathbb{E}^l)$, we say that $G^k \subseteq G^l$ if for all $V_i \rightarrow V_j \in \mathbb{E}^k$ exists an edge between $V_i$ and $V_j$ in $G^l$ such that $V_i \rightarrow V_j \in \mathbb{E}^k$ or $V_i - V_j \in \mathbb{E}^l$.

For the remainder of the paper, any inclusion between graphs will be intended as per Definition 3.2.

The masked server graph based on the $k$-th client graph, denoted as $\mu(\hat{G}, \mathcal{C}(G_{\Phi^k}))$, will be equal to the client graph, $\mu(\hat{G}, \mathcal{C}(G_{\Phi^k})) = \mathcal{C}(G_{\Phi^k})$, when the server graph converges to the true CPDAG, if Assumption 2.2 holds. Ultimately, the scoring function will be minimal when each client graph is included in the server one, as per Definition 3.2.

The regret function can be rewritten as:

$$R_k(H) = L(\mu(H, \mathcal{C}(G_{\Phi^k})), \mathbb{D}^k) - L(\mathcal{C}(G_{\Phi^k}), \mathbb{D}^k). \quad (3)$$

Incorporating Equation (3) into the optimization problem of Equation (2) allows the discovery of a representative of the MEC of the true causal DAG, for which all client-level CPDAGs are subgraphs.

**Theorem 3.3.** *Let $G$ denote the true server causal DAG, and let $\mathcal{C}(G)$ denote its corresponding CPDAG. Let $\Phi$ denote the family of unknown intervention targets across the $K$ clients. For each client $k \in \{1, \ldots, K\}$, let us denote the number of client samples with $n^k$ and the client-specific causal DAG as $G_{\Phi^k}$. Let $\hat{G}$ be defined as the graph approximated by solving Equation (2) using the regret function in Equation (3). If at the client-level, the CPDAG $\mathcal{C}(G_{\Phi^k})$ of each $G_{\Phi^k}$ is known to client $k$, $L$ is a consistent scoring function, and Assumption 2.2 holds, then $\hat{G}$ converges to $\mathcal{C}(G)$ for $n^1, \ldots, n^K \rightarrow \infty$.*

We emphasize that causal sufficiency and faithfulness are not explicitly assumed in the statement of the theorem, since

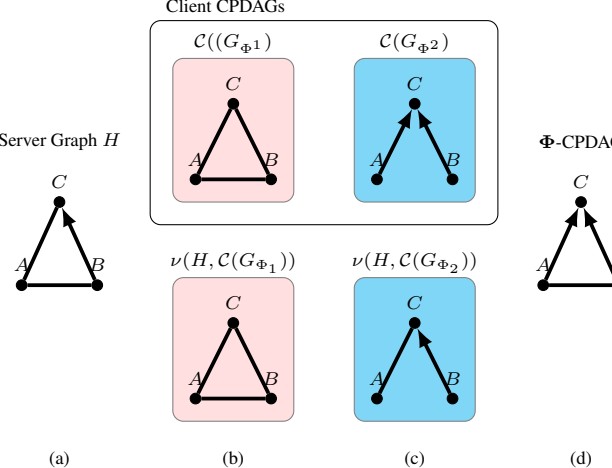

*Figure 3.* (a) Server graph obtained at an intermediate iteration (prior to convergence); (b) client 1 CPDAG (top) and undirected masking of the server CPDAG and client 1 CPDAG (bottom); (c) client 2 CPDAG (top) and undirected masking of the server CPDAG and client 2 CPDAG (bottom); (d) target $\Phi$-CPDAG result of the second phase of I-PERI.

the client-level CPDAGs are assumed to be available. When this assumption does not hold, each client CPDAG can be estimated from data under the standard assumptions of causal sufficiency and faithfulness using classical causal discovery methods, such as the PC or GES algorithms.

### 3.2. Orientation Refinement

In the presence of structural interventions, additional v-structures may become identifiable, as shown in Figure 1 (c). A structural intervention transforming a shielded collider into an unshielded one enables orienting the corresponding edges in the server graph. The second phase of I-PERI refines the orientation of ambiguous (i.e., undirected) edges in the global CPDAG based on client-level information. To this end, we penalize edges that are unoriented in the server CPDAG but are oriented in a client graph. This is achieved by computing the regret over the masking between the server graph —derived from the CPDAG obtained in the previous step —and the client graph, Definition 3.4, where undirected edges take precedence over directed ones (Figure 3). Consequently, edges that have been structurally intervened upon at the client-level are removed from the mask, while undirected edges in the server graph are left unoriented in the masked graph.

**Definition 3.4** (Undirected-consensus masking). Given two CPDAGs, $G^k = (\mathbb{V}, \mathbb{E}^k)$ and $G^l = (\mathbb{V}, \mathbb{E}^l)$, the undirected-consensus masking between the two graphs is defined as follows:

$$\nu(G^k, G^l) = (\mathbb{V}, \mathbb{E}^\nu),$$

where $\mathbb{E}^\nu$ is defined as:

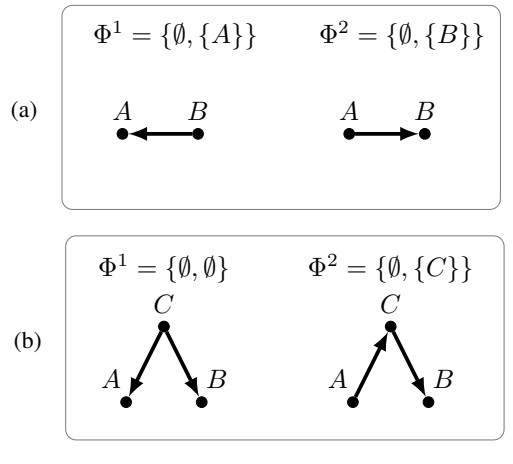

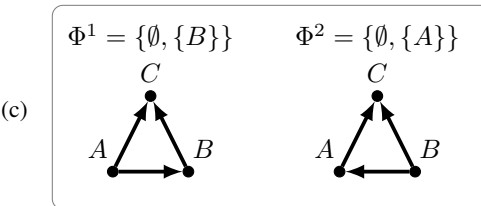

*Figure 4.* Three sets of server DAGs with their associated intervention targets where each set contains server DAGs that are $\Phi$-Markov equivalent.

- if $V_i \multimap V_j \in \mathbb{E}^k$ and $V_i \multimap V_j \in \mathbb{E}^l$, then $V_i \multimap V_j \in \mathbb{E}^\nu$.

- if $V_i \not\to\circ V_j$ in either $\mathbb{E}^k$ or $\mathbb{E}^l$, then $V_i \not\to\circ V_j \in \mathbb{E}^\nu$.

- if $V_i \to V_j \in \mathbb{E}^k$ and $V_i - V_j \in \mathbb{E}^l$, or vice versa, then $V_i - V_j \in \mathbb{E}^\nu$.

where $\multimap$ denotes an undirected edge, $-$, or a directed one, $\to$.

Since the regret function is based on a consistent and decomposable scoring function, it accounts for both possible orientations of undirected edges. Minimizing the regret requires orienting edges in the server graph according to the client graph. If no additional orientations can be inferred, e.g., in presence of observational data or parametric interventions, the regret will not decrease further, and the server graph will remain unchanged. We can thus rewrite Equation (1) as:

$$R_k(H) = L(\nu(H, \mathcal{C}(G_{\Phi^k})), \mathbb{D}^k) - L(\mathcal{C}(G_{\Phi^k}), \mathbb{D}^k). \quad (4)$$

The optimization problem remains the same as in Equation (2), but the search space is now limited to partially directed graphs derived from the server CPDAG by orienting undirected edges.

**$\Phi$-Markov Equivalence Class.** The specific setting of our federated problem does not explicitly provide the intervention targets; these cannot be shared and are possibly unknown by the clients themselves. Thus, we cannot use intervention targets, nor infer them, to identify the $\mathcal{I}$-MEC (Hauser & Bühlmann, 2012). However, the second step of I-PERI allows us to identify a tighter equivalence class than the observational MEC —but looser than the $\mathcal{I}$-MEC, as shown in the first example of Figure 4. Every additional orientation identified in the CPDAG at the client-level can be oriented in the global graph by using Equation (4) and the global CPDAG. Let us consider a family of unknown interventions across $K$ clients, $\boldsymbol{\Phi}$, observed only locally in a federated setting. We can then define the $\Phi$-MEC as follows:

**Definition 3.5** ($\Phi$-Markov Equivalence). Given two server graphs, $G^1$ and $G^2$, and relative intervention targets families, $\boldsymbol{\Phi_1}$ and $\boldsymbol{\Phi_2}$, we say that $G^1$ and $G^2$ are $\Phi$-Markov equivalent, given $\boldsymbol{\Phi_1}$ and $\boldsymbol{\Phi_2}$, denoted as $(G^1, \boldsymbol{\Phi_1}) \sim_{\boldsymbol{\Phi}} (G^2, \boldsymbol{\Phi_2})$, if and only if, for every disjoint sets $\mathbb{X}, \mathbb{Y}, \mathbb{Z} \subset \mathbb{V}$:

- $\mathbb{X} \perp\!\!\!\perp_{G^1} \mathbb{Y}|\mathbb{Z} \Leftrightarrow \mathbb{X} \perp\!\!\!\perp_{G^2} \mathbb{Y}|\mathbb{Z}$.

- $\exists \Phi_1^k \in \boldsymbol{\Phi_1}, \exists W \in \mathbb{V}\backslash\mathbb{X} \cup \mathbb{Y} \cup \mathbb{Z}, (\mathbb{X} \perp\!\!\!\perp_{G^1_{\Phi_1^k}} \mathbb{Y}|\mathbb{Z}) \wedge (\mathbb{X} \not\perp\!\!\!\perp_{G^1_{\Phi_1^k}} \mathbb{Y}|\mathbb{Z}, W) \Leftrightarrow \exists \Phi_2^l \in \boldsymbol{\Phi_2}, \exists W \in \mathbb{V}\backslash\mathbb{X} \cup \mathbb{Y} \cup \mathbb{Z}, (\mathbb{X} \perp\!\!\!\perp_{G^2_{\Phi_2^l}} \mathbb{Y}|\mathbb{Z}) \wedge (\mathbb{X} \not\perp\!\!\!\perp_{G^2_{\Phi_2^l}} \mathbb{Y}|\mathbb{Z}, W)$.

It is important to note that the sets $\Phi_1^k$ and $\Phi_2^l$ appearing in the definition are not required to correspond to the same intervention targets. For illustration, Figure 4, presents three sets of graphs, each consisting of graphs that are $\Phi$-Markov equivalent. Notably, within each class, the graphs are associated with different intervention target sets, despite being $\Phi$-Markov equivalent. Moreover, the definition does not require that the intervened graphs share the same skeleton. Figure 4 (b), the second graph given the intervention family $\boldsymbol{\Phi_2}$, will produce two mutilated graphs with different skeletons. In the following, we show that all graphs that are $\Phi$-Markov equivalent can be characterized graphically.

**Theorem 3.6** (Characterization of $\Phi$-Markov Equivalence Class). *Two server graphs $G^1$ and $G^2$ with unknown intervention targets families $\boldsymbol{\Phi_1}$ and $\boldsymbol{\Phi_2}$ belong to the same $\Phi$-Markov equivalence class ($\Phi$-MEC), $(G^1, \boldsymbol{\Phi_1}) \sim_{\boldsymbol{\Phi}} (G^2, \boldsymbol{\Phi_2})$, if and only if:*

1. *$G^1$ and $G^2$ have the same skeleton*

2. *$G^1$ and $G^2$ have the same v-structures*

3. *There exists $\Phi_1^k \in \boldsymbol{\Phi_1}$ such that a v-structure appears in $G^1_{\Phi_1^k}$ that is not present in $G^1$ if and only if there exists $\Phi_2^l \in \boldsymbol{\Phi_2}$ such that the same v-structure appears in $G^2_{\Phi_2^l}$ and is not present in $G^2$.*

| True DAG | CPDAG | $\Phi$-CPDAG | $\mathcal{I}$-CPDAG |
|----------|-------|--------------|---------------------|

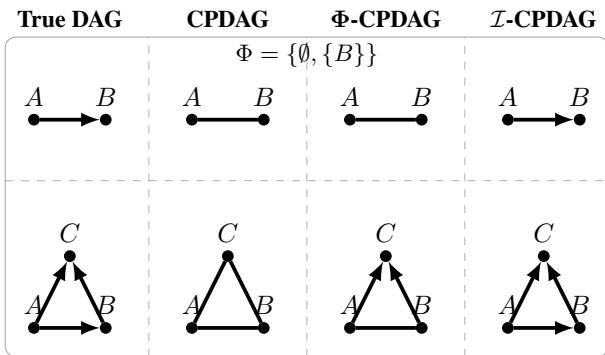

*Figure 5.* Example showing the difference between observational CPDAG, $\Phi$-CPDAG, and $\mathcal{I}$-CPDAG for the intervention family $\Phi = \{\emptyset, \{B\}\}$.

We now define the representative of a $\Phi$-MEC, denoted as $\Phi$-CPDAG, which is obtained from CPDAG by orienting part of its undirected edges.

**Definition 3.7** ($\Phi$-CPDAG). Let $G$ be a server causal DAG and $\Phi$ a set of intervention targets over multiple clients. Let us define the CPDAG of $G$ as $\mathcal{C}(G)$. The partially directed graph obtained by orienting in $\mathcal{C}(G)$ every edge oriented in the CPDAG of the intervened graph $\mathcal{C}(G_{\Phi^k})$ for every $\Phi^k \in \Phi$ is called the $\Phi$-CPDAG of $G$. We will denote such a graph as $\Phi(G)$.

Given a $\Phi$-Markov equivalence class, the $\Phi$-CPDAG for that class is unique. We emphasize this result with a corollary.

**Corollary 3.8.** *Let $\Phi_1(G^1)$ and $\Phi_2(G^2)$ be the $\Phi$-CPDAGs of two causal DAGs $G^1$ and $G^2$ and $\Phi_1$ and $\Phi_2$ two intervention families with unknown targets. If ($\Phi$-MEC), $(G^1, \Phi_1) \sim_\Phi (G^2, \Phi_2)$, then $\Phi_1(G^1) = \Phi_2(G^2)$.*

We want to highlight that the $\Phi$-CPDAG is a representative of the $\Phi$-MEC that can be tighter than the observational CPDAG, while remaining looser than the $\mathcal{I}$-CPDAG. As illustrated in Figure 5, in the simple two-node case where one node causes the other, interventions do not provide enough information in a federated setting to further refine the observational CPDAG. However, when an intervention reveals a v-structure, additional edges in the observational CPDAG can be oriented, yielding a tighter characterization of the $\Phi$-MEC.

We can now show that I-PERI converges to the $\Phi$-CPDAG of the true causal DAG.

**Theorem 3.9** (Correctness of I-PERI). *Let $G$ denote the true server causal DAG, and let $\mathcal{C}(G)$ denote its corresponding CPDAG. Let $\Phi$ denote the family of unknown intervention targets across the $K$ clients. For each client $k \in \{1, \ldots, K\}$, let $G_{\Phi^k}$ denote the client-specific causal DAG. Let $\hat{G}$ be the output of I-PERI. If at the client-level,*

*the CPDAG $\mathcal{C}(G_{\Phi^k})$ of each $G_{\Phi^k}$ is known to client $k$, $L$ is a consistent scoring function, and Assumption 2.2 holds, then $\hat{G}$ converges to $\Phi(G)$ for $n^1, \ldots, n^K \to \infty$.*

As in the case of Theorem 3.3, causal sufficiency and faithfulness are not stated explicitly in Theorem 3.9, as the result is formulated under the premise that the client-level CPDAGs are available. When these are unknown, they can be estimated using PC or GES under the omitted assumptions.

### 3.3. Privacy Guarantees

Sharing regrets between clients and the server reveals less information than sharing local graphs or model parameters. Nonetheless, sharing the global causal graph may still violate privacy requirements, although this risk can be mitigated using standard encryption techniques. Moreover, given the shared regrets and the global graph, one could in principle reconstruct each client's local causal graph using Equation (2), (Mian et al., 2023). While this would enable identification of client-level interventions, the reconstruction problem is NP-hard (Chickering et al., 2004). We can provide formal privacy guarantees for I-PERI by enforcing $\epsilon$-differential privacy via the Laplace mechanism (Dwork, 2006). Differential privacy ensures that the algorithm's output changes only marginally when a single individual's data is added or removed. Formally, for any two datasets $\mathbb{D}^k$ and $\mathbb{D}'^k$ differing in one record, and for any output event, the ratio of the corresponding probabilities is bounded by $e^\epsilon$, with smaller $\epsilon$ implying stronger privacy. The Laplace mechanism achieves $\epsilon$-differential privacy for numeric queries by adding zero-mean Laplace noise with scale proportional to the query's global sensitivity divided by $\epsilon$, thereby masking individual contributions while preserving utility. Let us consider that each graph at the local level, $G^k$, is described by a set of parameters $\theta^k$. Assuming to have a consistent scoring function $L$, and that $||\theta^k - \theta'^k|| \propto \frac{1}{n}$ (Mian et al., 2023) we can bound the sensitivity of the regret function.

**Lemma 3.10.** *Assume $P_k(x; \theta)$ to be uniformly lower-bounded by $r$, namely, $\forall x \in \mathbb{D} \; \forall \theta \in \theta : P_k(x, \theta) \geq r$, that $||\theta|| \leq M$ for all model parameters $\theta \in \Theta$ and that the score $L$ is partially differentiable with respect to $\theta$. Let $\mathbb{D}_k$ and $\mathbb{D}'_k$ be datasets that differ in a single element, $\mathbb{D}_k \backslash \mathbb{D}'_k = x_i$, and that $\theta$ and $\theta'$ are the respective local parameters, with respective regrets $\hat{R}_k(G)$ and $\hat{R}'_k(G)$. We assume that $||\theta - \theta'|| \leq \frac{2M}{n}$. Then the sensitivity of the regret function is bounded by[2]:*

$$\max_k |\hat{R}_k(G) - \hat{R}'_k(G)| \leq (2M + 1) \log r^2 + \mathcal{O}\left(\frac{\log n}{n}\right).$$

We can then add Laplacian noise to the regret function

---

[2] Note that this bounding corrects a minor mistake in the original proof of Mian et al. (2023).

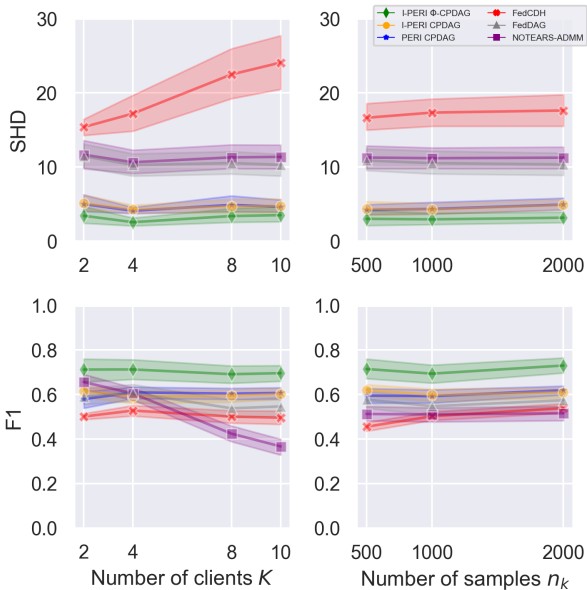

*Figure 6.* Results on linear synthetic data using PC to discover the client CPDAG. The plot compares SHD ($\downarrow$) and F1 score ($\uparrow$) across different baselines. On the left side, with an increasing number of clients $K$, while on the right side, with an increasing number of per-client samples $n$. Error bars represent confidence intervals.

before transmission to the server.

**Proposition 3.11.** *Assume each local regret $\hat{R}_k$ has sensitivity $\leq Q$, then I-PERI with i.i.d. Laplace noise with scale $\lambda = \frac{Q}{\epsilon}$ added to each $\hat{R}_k$ is $\epsilon$-differentially private.*

This generalizes to the regret in Equation (3) and Equation (4) since the intersection is operated at the client-level.

## 4. Experiments

We evaluate I-PERI on linear synthetic datasets[3] and compare it against state-of-the-art methods for FCD, namely PERI (Mian et al., 2023), FedDAG (Gao et al., 2023), NOTEARS-ANMM (Ng & Zhang, 2022), and FedCDH (Li et al., 2024). Baselines are selected based on code availability. We consider multiple experimental settings: (i) varying the number of variables ($|\mathbb{V}| \in \{3, 4, 8, 10, 20\}$), (ii) the number of clients ($K \in \{2, 4, 8, 10\}$), (iii) the per-client sample size ($n_k \in \{500, 1000, 2000\}$), and (iv) heterogeneous sample sizes across clients. Performance is evaluated using Structural Hamming Distance (SHD) and F1 score with respect to the true DAG. Results are reported in Figure 6 and Table 1.

The true DAGs are generated following the Erdos-Renyi model (Erdös & Rényi, 1960), with an expected num-

---

[3] https://github.com/CIPHOD/pyCIPHOD/tree/main/reproducibility/icml2026

ber of edges equal to the number of nodes. Each client dataset is generated according to a linear structural equation model with additive Gaussian noise of the form $V_i = \sum_{V_j \in Pa_i^G} w_{ji}V_j + N_i$, where the noise $N_i \sim \mathcal{N}(0, 1)$ and the edge weights $w_{ji}$ are sampled uniformly at random from the union of the intervals $[-1, -0.1]$ and $[0.1, 1]$. To highlight the improvements introduced by I-PERI in its second phase, we artificially introduce shielded colliders and prioritize client-level interventions that create v-structures not identifiable from purely observational data. Each client holds data generated under a single intervention on the true DAG, except for one holding exclusively observational data, per Assumption 2.2. For PERI and I-PERI, client-level CPDAGs are learned using PC (results with GES are reported in Appendix B), and regrets are computed using the BIC score. All results are averaged over 10 random seeds, chosen to ensure a client-level CPDAG F1 score above 0.85, which is necessary for reliable downstream orientation. Experiments varying the number of clients are averaged over both homogeneous and heterogeneous sample sizes; in the latter case, clients are randomly assigned 500, 1000, or 2000 samples. NOTEARS-ANMM is excluded from heterogeneous settings due to its requirement for equal sample sizes. Overall, the results in Figure 6 confirm the effectiveness of I-PERI in orienting additional edges beyond those identified in the initial CPDAG. Performance is consistent across most configurations in Figure 6; this holds for experiments in which GES is used as local discovery method, as shown in Appendix B (a). Notably, I-PERI consistently outperforms all baselines across all settings and exhibits very narrow confidence intervals. Table 1 reports performance as the number of variables increases. As shown in the second row, I-PERI achieves the best performance in most cases, with the exception of the 10-variable setting, where it ranks second. More generally, I-PERI consistently outperforms the baselines, with only a few cases in which its performance is comparable to the strongest competing method. Finally, Figure 7 highlights the computational efficiency of I-PERI, which achieves substantially lower —by orders of magnitude —runtime than competing methods. Note that I-PERI does not assume linearity and can be applied to non-linear datasets. Its ability to handle non-linear data depends on the causal discovery method employed at the client-level. In Appendix B, we report results on non-linear synthetic data using PC as the local discovery method, confirming I-PERI's effectiveness in this setting.

## 5. Discussion and Conclusion

This work addresses a key limitation of existing FCD methods by relaxing the common assumption that all clients share an identical causal model and experience no interventions. By explicitly accounting for unknown client-level interventions, our approach better reflects the heterogeneity

*Table 1.* Average SHD and F1 score $\pm$ standard deviation of I-PERI and baselines across 10 random seeds for varying number of variables $p$. Best results are highlighted in gray.

| Method | Metric | $p = 3$ | $p = 4$ | $p = 8$ | $p = 10$ | $p = 20$ |
|---|---|---|---|---|---|---|
| PERI | **SHD** | $3.16 \pm 0.55$ | $4.43 \pm 1.13$ | $8.40 \pm 0.51$ | $11.75 \pm 2.21$ | $30.0 \pm 10.02$ |
| | **F1** | $0.59 \pm 0.19$ | $0.62 \pm 0.09$ | $0.64 \pm 0.04$ | $0.61 \pm 0.06$ | $0.56 \pm 0.12$ |
| I-PERI | **SHD** | $1.53 \pm 1.16$ | $2.87 \pm 1.88$ | $4.44 \pm 3.04$ | $9.85 \pm 3.43$ | $27.8 \pm 4.79$ |
| | **F1** | $0.75 \pm 0.23$ | $0.69 \pm 0.19$ | $0.74 \pm 0.16$ | $0.58 \pm 0.16$ | $0.51 \pm 0.07$ |
| NOTEARS-ADMM | **SHD** | $1.64 \pm 1.06$ | $2.99 \pm 1.44$ | $8.44 \pm 2.55$ | $13.70 \pm 3.27$ | $29.45 \pm 6.29$ |
| | **F1** | $0.54 \pm 0.35$ | $0.55 \pm 0.29$ | $0.46 \pm 0.19$ | $0.50 \pm 0.12$ | $0.49 \pm 0.09$ |
| FedDAG | **SHD** | $3.01 \pm 1.89$ | $3.46 \pm 2.39$ | $6.68 \pm 2.61$ | $9.04 \pm 4.52$ | $30.74 \pm 5.55$ |
| | **F1** | $0.49 \pm 0.31$ | $0.63 \pm 0.27$ | $0.72 \pm 0.10$ | $0.75 \pm 0.12$ | $0.22 \pm 0.13$ |
| FedCDH | **SHD** | $2.27 \pm 1.17$ | $4.83 \pm 1.74$ | $14.86 \pm 4.24$ | $25.97 \pm 5.33$ | $61.74 \pm 13.35$ |
| | **F1** | $0.68 \pm 0.16$ | $0.58 \pm 0.16$ | $0.44 \pm 0.13$ | $0.36 \pm 0.09$ | $0.28 \pm 0.06$ |

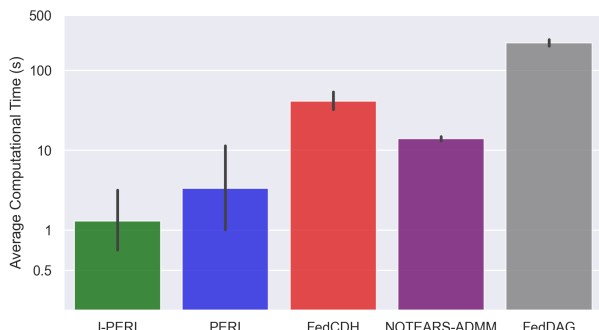

*Figure 7.* Symlog computational time in seconds averaged over all experiments for I-PERI and each baseline method.

encountered in real-world federated environments, such as multi-center clinical studies or distributed organizational data systems. A central insight of this paper is that client heterogeneity induced by interventions can be leveraged rather than ignored. While such heterogeneity is often viewed as a complication in federated learning, I-PERI exploits intervention-induced structural differences across clients to orient additional edges beyond what is identifiable from purely observational data. This results in a strictly tighter equivalence class, captured by the proposed $\Phi$-MEC. Furthermore, I-PERI enforces privacy by design, through restricted information sharing and server-side aggregation. This bottom-up approach offers a favorable trade-off between privacy, computational efficiency, and statistical power, though it may not provide the same formal guarantees as cryptographic methods in adversarial settings. In the main body of this work, we focused on structural interventions; nevertheless, I-PERI is also sound in settings with purely observational data or parametric interventions only. When client data contains exclusively parametric interventions or no interventions at all, I-PERI reduces to recovering the CPDAG common to all clients. In this case, parametric interventions and purely observational data are treated equivalently by the algorithm, as they do not alter

the underlying causal structure. Consequently, no additional edge orientations beyond those identifiable from observational data can be inferred at the server-level. The intuition is straightforward: if one client holds purely observational data while another client applies a parametric intervention, both datasets are generated from the same causal DAG, differing only in the functional parameters or noise distributions. As a result, each client can locally identify the same equivalence class, and aggregating this information at the server-level yields the standard CPDAG of the underlying DAG. Despite these advantages, this work has several limitations. The accuracy of I-PERI strongly hinges on the accuracy of the CPDAG learned at the client-level. If clients cannot reliably recover their local CPDAGs, the performance of I-PERI may degrade. Any additional erroneous orientation will be reflected in the final $\Phi$-CPDAG learned at the server-level. Moreover, our theoretical analysis relies on standard assumptions in causal discovery, including causal sufficiency, no selection bias, and faithfulness. These assumptions may be violated in real-world settings. Extending the proposed framework to settings with latent variables, selection bias, or weaker assumptions remains an important direction for future research.

## Acknowledgements

This work was supported by the CIPHOD project (ANR-23-CPJ1-0212-01). We thank Osman Mian, Michael Kamp, and Simon Ferreira for their assistance with the differential privacy proofs. We also thank Sarah Semlali for helping revise the final version of this paper.

## Impact Statement

This paper presents work whose goal is to advance the field of Federated Causal Discovery. By enabling causal discovery across decentralized institutions without requiring direct data sharing, this work has potential benefits in privacy-sensitive domains such as healthcare. There are no direct

negative societal consequences we feel must be specifically highlighted here.

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

# A. Proofs

**Theorem 3.3.** *Let $G$ denote the true server causal DAG, and let $\mathcal{C}(G)$ denote its corresponding CPDAG. Let $\Phi$ denote the family of unknown intervention targets across the $K$ clients. For each client $k \in \{1, \ldots, K\}$, let us denote the number of client samples with $n^k$ and the client-specific causal DAG as $G_{\Phi^k}$. Let $\hat{G}$ be defined as the graph approximated by solving Equation (2) using the regret function in Equation (3). If at the client-level, the CPDAG $\mathcal{C}(G_{\Phi^k})$ of each $G_{\Phi^k}$ is known to client $k$, $L$ is a consistent scoring function, and Assumption 2.2 holds, then $\hat{G}$ converges to $\mathcal{C}(G)$ for $n^1, \ldots, n^K \to \infty$.*

*Proof.* Per Theorem 1 and Corollary 3 of Mian et al. (2023), assuming that all clients share the same graph, for $n^1, \ldots, n^K \to \infty$ than $\hat{G}$ converges to $\mathcal{C}(G)$. The regret functional form in Equation (3) is the same as the original one in Equation (1). The only change is on the graph to which we compare the client graph when computing the regret. In our setting, each client graph $G_{\Phi^k}$ is derived from the true graph $G$ via structural interventions $\Phi^k$. More specifically, we compute the regret over $\mu(\hat{G}, \mathcal{C}(G_{\Phi^k}))$, per Definition 3.1. In this case, the regret function in Equation (3) will be minimal when $\mu(\hat{G}, \mathcal{C}(G_{\Phi^k})) = \mathcal{C}(G_{\Phi^k})$, meaning that $\mathcal{C}(G_{\Phi^k}) \subseteq \hat{G}$. Additionally, only edges that lead to improvements in the score will be added. Since we start from an empty graph, we will never add edges that are not in $C(G)$, or in mutilated graphs derived from it, $C(G_{\Phi^k})$, meaning that $\hat{G} \subseteq \mathcal{C}(G)$. Per Assumption 2.2, at least one client $k'$ is purely observational, i.e., $\Phi^{k'} = \emptyset$, implying that for this client, $\mathcal{C}(G_{\Phi^{k'}}) = \mathcal{C}(G)$, and $\mu(\hat{G}, \mathcal{C}(G)) = \mathcal{C}(G)$, that is to say $\mathcal{C}(G) = \hat{G}$. $\qquad\square$

**Theorem 3.6** (Characterization of $\Phi$-Markov Equivalence Class). *Two server graphs $G^1$ and $G^2$ with unknown intervention targets families $\Phi_1$ and $\Phi_2$ belong to the same $\Phi$-Markov equivalence class ($\Phi$-MEC), $(G^1, \Phi_1) \sim_{\Phi} (G^2, \Phi_2)$, if and only if:*

1. *$G^1$ and $G^2$ have the same skeleton*

2. *$G^1$ and $G^2$ have the same v-structures*

3. *There exists $\Phi_1^k \in \Phi_1$ such that a v-structure appears in $G_{\Phi_1^k}^1$ that is not present in $G^1$ if and only if there exists $\Phi_2^l \in \Phi_2$ such that the same v-structure appears in $G_{\Phi_2^l}^2$ and is not present in $G^2$.*

*Proof.* ($\Rightarrow$) If $(G^1, \Phi_1) \sim_{\Phi} (G^2, \Phi_2)$ by Definition 3.5 they imply the same observational $d$-separations. Hence, they have the same skeleton and v-structures (Verma & Pearl, 1990). Assuming that $(G^1, \Phi_1) \sim_{\Phi} (G^2, \Phi_2)$ does not imply the third condition, meaning that there are not two $\Phi_1^k$ and $\Phi_2^l$ such that if $G_{\Phi_1^k}^1$ has a v-structure the same is present in $G_{\Phi_2^l}^2$. This directly violates the second condition in Definition 3.5.

($\Leftarrow$) If the two graphs have the same skeleton and v-structures, they have the same observational $d$-separations, satisfying the first condition of Definition 3.5. Condition three ensures that every v-structure appearing in one graph under an intervention, $G_{\Phi_1^k}^1$, implies that there exists another intervention in the second graph, $G_{\Phi_2^l}^2$, that generates the same v-structure. Thus, the second condition of Definition 3.5 is satisfied, meaning that $(G^1, \Phi_1) \sim_{\Phi} (G^2, \Phi_2)$. $\qquad\square$

**Corollary A.1.** *Let $\Phi_1(G^1)$ and $\Phi_2(G^2)$ be the $\Phi$-CPDAGs of two causal DAGs $G^1$ and $G^2$ and $\Phi_1$ and $\Phi_2$ two intervention families with unknown targets. If ($\Phi$-MEC), $(G^1, \Phi_1) \sim_{\Phi} (G^2, \Phi_2)$, then $\Phi_1(G^1) = \Phi_2(G^2)$.*

*Proof.* From Theorem 3.6 and Definition 3.7 $\qquad\square$

**Theorem 3.9** (Correctness of I-PERI). *Let $G$ denote the true server causal DAG, and let $\mathcal{C}(G)$ denote its corresponding CPDAG. Let $\Phi$ denote the family of unknown intervention targets across the $K$ clients. For each client $k \in \{1, \ldots, K\}$, let $G_{\Phi^k}$ denote the client-specific causal DAG. Let $\hat{G}$ be the output of I-PERI. If at the client-level, the CPDAG $\mathcal{C}(G_{\Phi^k})$ of each $G_{\Phi^k}$ is known to client $k$, $L$ is a consistent scoring function, and Assumption 2.2 holds, then $\hat{G}$ converges to $\Phi(G)$ for $n^1, \ldots, n^K \to \infty$.*

*Proof.* From Theorem 3.3, I-PERI converges to $\hat{G}$, which is the CPDAG $\mathcal{C}(G)$ of the true graph $G$. The regret functional in Equation (4) has the same form as the original one in Equation (1); the only difference lies in the reference graph used to compute the regret. Specifically, when evaluating the regret, we remove from the server graph all edges that are absent

in the client graph. Equivalently, the regret is computed over $\nu(\hat{G}, \mathcal{C}(G_{\Phi^k}))$, as per Definition 3.1. The regret function in Equation (4) is minimized when

$$\nu(\hat{G}, \mathcal{C}(G_{\Phi^k})) = \mathcal{C}(G_{\Phi^k}).$$

In this case, for every node $V_i$, the parent set in the client graph coincides with the corresponding parent set in the restricted server graph:

$$L\big(V_i, \mathrm{Pa}_{G_{\Phi^k}}(V_i); \mathbb{D}^k\big) = L\Big(V_i, \mathrm{Pa}_{\nu(\hat{G}, \mathcal{C}(G_{\Phi^k}))}(V_i); \mathbb{D}^m\Big),$$

which implies

$$\mathrm{Pa}_{G_{\Phi^k}}(V_i) = \mathrm{Pa}_{\nu(\hat{G}, \mathcal{C}(G_{\Phi^k}))}(V_i).$$

Therefore, minimizing the regret requires orienting all edges that are oriented in the client graph. These orientations typically arise from client-level interventions that reveal new v-structures. Starting from the CPDAG common to all client CPDAGs, this process amounts to orienting previously undirected edges based on v-structures identified at the client-level. By Definition 3.7, this corresponds to recovering the $\Phi$-CPDAG of the true graph $G$. □

**Lemma A.2.** *Assume $P_k(x; \theta)$ to be uniformly lower-bounded by $r$, namely, $\forall x \in \mathbb{D} \,\forall \theta \in \theta : P_k(x, \theta) \geq r$, that $||\theta|| \leq M$ for all model parameters $\theta \in \Theta$ and that the score $L$ is partially differentiable with respect to $\theta$. Let $\mathbb{D}_k$ and $\mathbb{D}'_k$ be datasets that differ in a single element, $\mathbb{D}_k \backslash \mathbb{D}'_k = x_i$, and that $\theta$ and $\theta'$ are the respective local parameters, with respective regrets $\hat{R}_k(G)$ and $\hat{R}'_k(G)$. We assume that $||\theta - \theta'|| \leq \frac{2M}{n}$. Then the sensitivity of the regret function is bounded by[4]:*

$$\max_k |\hat{R}_k(G) - \hat{R}'_k(G)| \leq (2M + 1)\log r^2 + \mathcal{O}\Big(\frac{\log n}{n}\Big).$$

*Proof.* Following the proof to Lemma 4 in (Mian et al., 2023), we provide a more refined proof that corrects a minor mistake in the original paper. Removing a sample from $\mathbb{D}_k$ also changes the local DAG, $G^k$. For simplicity, we will represent the graph as a set of parameters, $\theta^k$. We can thus write the sensitivity of the scoring function as follows:

$$\max_k |\hat{R}_k(G) - \hat{R}'_k(G)| = \max_k |L(\mathbb{D}_k, \theta) - L(\mathbb{D}_k, \theta^k) -$$
$$- L(\mathbb{D}'_k, \theta') + L(\mathbb{D}'_k, \theta'^k)|$$
$$\leq \max_k |L(\mathbb{D}_k, \theta') - L(\mathbb{D}'_k, G)| +$$
$$+ |L(\mathbb{D}'_k, \theta'^k) - \mathbb{D}_k, \theta^k)|$$

We can bound the absolute values above, which differ in only one element and model parameters. We have that:

$$|L(\mathbb{D}, \theta) - L(\mathbb{D}', \theta')| \leq |L(\mathbb{D}', \theta) - L(\mathbb{D}, \theta) +$$
$$+ ||\theta - \theta'|| \, |L(\mathbb{D}, \theta') - L(\mathbb{D}, \theta')|$$

Meaning that we can bound the absolute values in the sensitivity:

$$\max_k |\hat{R}_k(G) - \hat{R}'_k(G)| = \underbrace{|L(x_k, \theta)|}_{\leq \log r}$$
$$+ \underbrace{||\theta - \theta'||}_{\leq 2M/n} |L(\mathbb{D}_k, \theta')L(\mathbb{D}_k, \theta)|$$
$$+ |L(x_k, \theta^k)| + ||\theta^k - \theta'^k|| \underbrace{|L(\mathbb{D}_k, \theta') - L(\mathbb{D}_k, \theta)|}_{\propto n} + \mathcal{O}\Big(\frac{\log n}{n}\Big)$$
$$\leq \log r + 2M\log r + \log r + 2M\log r + \mathcal{O}\Big(\frac{\log n}{n}\Big)$$
$$= (2M + 1)\log r^2 + \mathcal{O}\Big(\frac{\log n}{n}\Big)$$

---

[4]Note that this bounding corrects a minor mistake in the original proof of Mian et al. (2023).

The previous inequality was:

$$\max_k |\hat{R}_k(G) - \hat{R}'_k(G)| \leq (4M + 1)\log r + \mathcal{O}\left(\frac{\log n}{n}\right)$$

$\square$

**Proposition A.3.** *Assume each local regret $\hat{R}_k$ has sensitivity $\leq Q$, then I-PERI with i.i.d. Laplace noise with scale $\lambda = \frac{Q}{\epsilon}$ added to each $\hat{R}_k$ is $\epsilon$-differentially private.*

*Proof.* Follows from Lemma 3.10 and Proposition 5 of (Mian et al., 2023). $\square$

# B. Experimental Results

Following additional experimental:

- Figure 8a shows the results on linear synthetic data using GES as client-level causal discovery method.

- Figure 8b compares the results using PC and GES as client-level causal discovery methods.

- Figure 9 reports the results on non-linear synthetic data using PC as client-level causal discovery method.

- Table 2 compares I-PERI and PERI on non-linear synthetic data as the number of variables increases.

Non-linear synthetic data was generated following the same procedure as in the linear case, but with non-linear transformations applied to the underlying variables. Specifically, the data generation process employed three functional forms: the hyperbolic tangent $\tanh(\cdot)$, the sine function $\sin(\cdot)$, and the quadratic function $x^2$, which were randomly selected to define the structural model equations. The resulting continuous data were then discretized into 5 uniform bins. Client-level causal discovery was subsequently performed using the PC algorithm with the $\chi^2$ conditional independence test.

*Table 2.* Average SHD and F1 score $\pm$ standard deviation of I-PERI and PERI across 10 random seeds for varying number of variables $p$ on non-linear synthetic data. Best results are highlighted in gray.

| Method | Metric | $p = 3$ | $p = 4$ | $p = 8$ | $p = 10$ | $p = 20$ |
|--------|--------|---------|---------|---------|----------|----------|
| PERI | **SHD** | $3.26 \pm 0.68$ | $4.36 \pm 0.92$ | $9.1 \pm 2.46$ | $13.75 \pm 3.33$ | $27.33 \pm 15.37$ |
| | **F1** | $0.57 \pm 0.22$ | $0.61 \pm 0.12$ | $0.59 \pm 0.11$ | $0.55 \pm 0.09$ | $0.58 \pm 0.17$ |
| I-PERI | **SHD** | $1.78 \pm 1.34$ | $4.0 \pm 1.80$ | $9.33 \pm 3.01$ | $10.53 \pm 3.90$ | $24.09 \pm 5.83$ |
| | **F1** | $0.71 \pm 0.27$ | $0.52 \pm 0.27$ | $0.50 \pm 0.16$ | $0.57 \pm 0.14$ | $0.56 \pm 0.08$ |

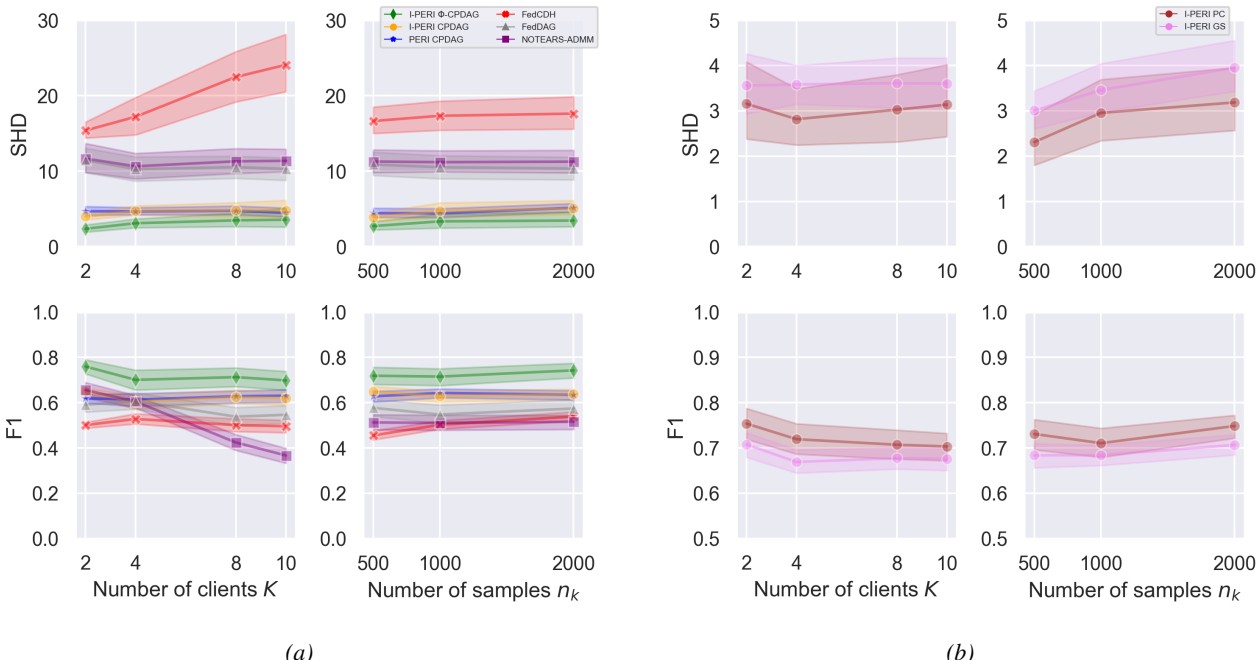

*(a)*                                          *(b)*

*Figure 8.* (a) Results on linear synthetic data using GES to discover the client CPDAG. The plot compares SHD ($\downarrow$), log scaled, and F1 score ($\uparrow$) across different baselines. On the left side, with an increasing number of clients $K$, while on the right side, with an increasing number of per-client samples $n$. Error bars represent confidence intervals. (b) Results comparing PC and GES as client-level causal discovery methods. The plot compares SHD ($\downarrow$) and F1 score ($\uparrow$) across different baselines. Error bars represent confidence intervals.

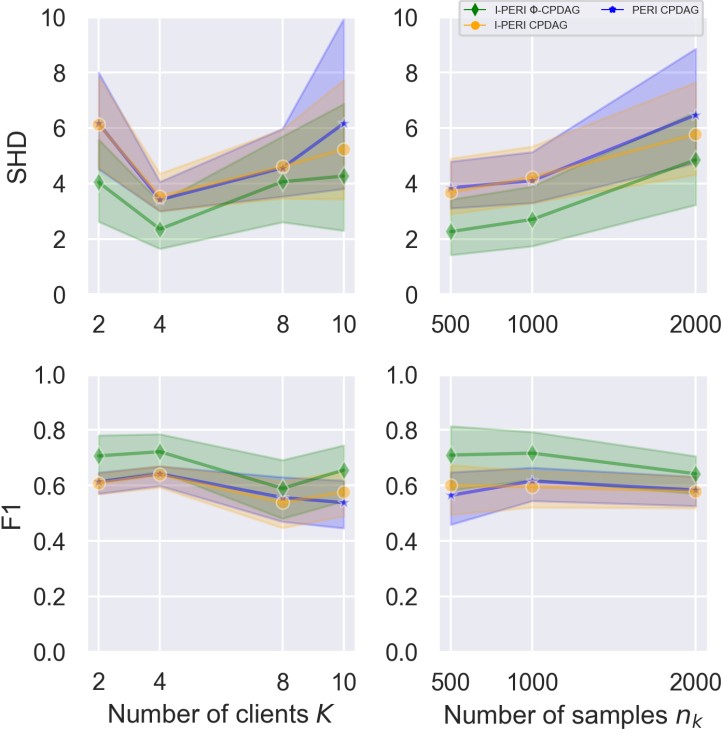

*Figure 9.* Results on non-linear synthetic data using PC to discover the client CPDAG. The plot compares SHD ($\downarrow$), log scaled, and F1 score ($\uparrow$) between I-PERI and PERI. On the left side, with an increasing number of clients $K$, while on the right side, with an increasing number of per-client samples $n$. Error bars represent confidence intervals.

