# OpenReview forum: "Regret-Based Federated Causal Discovery with Unknown Interventions"
_ICML.cc/2026/Conference — ICML 2026 regular_

### Official Review · Reviewer_tLLZ · 2026-03-12

**Soundness:** 2
**Presentation:** 3
**Significance:** 3
**Originality:** 3
**Overall Recommendation:** 4
**Confidence:** 2

**Summary:**

This paper studies federated causal discovery under unknown client-level interventions. Specifically, the authors propose I-PERI, a regret-based federated causal discovery algorithm with a two-stage strategy. First, the server recovers the global CPDAG by minimizing client regrets. Second, the algorithm exploits structural differences induced by interventions across clients to orient additional edges, resulting in a tighter equivalence class representation called the Θ-CPDAG. Experiments on synthetic datasets demonstrate that I-PERI consistently improves SHD and F1 score compared with several federated causal discovery baselines while maintaining good computational efficiency.

**Compliance With Llm Reviewing Policy:**

Affirmed.

**Key Questions For Authors:**

Please refer to the weaknesses above. In addition, I am not very familiar with the typical experimental conventions for this setting. Therefore, if using purely synthetic data is standard practice in this field, the authors may clarify this and disregard Weakness 1.

**Strengths And Weaknesses:**

Strengths:
1. The proposed two-stage framework (I-PERI) separates structure recovery and edge orientation. It first recovers the global CPDAG and then refines orientations using intervention-induced structural differences across clients. This design is intuitive and interpretable.
2. The authors provide theoretical guarantees, showing that I-PERI converges to the Θ-CPDAG under standard assumptions and large sample sizes, and also derive sensitivity bounds for regret to enable differential privacy.
3. The framework shares only regret values instead of raw data or local graphs, and can incorporate differential privacy via Laplace noise, offering a privacy-preserving mechanism suitable for federated settings.


Weaknesses:
1. All experiments are conducted on synthetic datasets, with no real-world datasets included. This limits the evaluation of the method's practical applicability.
2. The experiments involve relatively small graphs (up to 10 variables) and limited numbers of clients, leaving scalability to larger causal graphs largely unexplored.
3. The method strongly relies on the accuracy of local CPDAG estimation. Errors in client-level causal discovery may propagate to the federated stage and degrade the final Θ-CPDAG.

---

> ### Author Rebuttal · Authors · 2026-03-31
>
> We thank the Reviewer for taking the time to provide valuable feedback. Below, we provide a point-by-point reply to the concerns raised:
>
> Q: “All experiments are conducted on synthetic datasets…”
>
> A: We acknowledge the limitations of using only synthetic datasets. However, finding suitable real-world interventional datasets is particularly challenging; as in the case of [4] (see response to Reviewer W3JC), data are often unavailable due to privacy reasons. We tested FedDAG and I-PERI (restricting to these two methods as they explicitly handle heterogeneous client data sizes) on the Sachs dataset [6], which contains measurements of protein and phospho-protein levels in human immune system cells under various experimental conditions. The dataset includes six interventions (including no intervention, in compliance with Assumption 2.2), and we consider six clients, each holding data corresponding to a distinct intervention. The overall performance of all methods is quite poor (FedDAG: SHD=21, F1=0.22; PERI: SHD=27, F1=0.26; I-PERI: SHD=23, F1=0.29). We are currently searching for real-world datasets that could yield more insightful results, and are working on the IT monitoring data used in [2] (see response to Reviewer W3JC).
>
> Q: “The experiments involve relatively small graphs…”
>
> A: We acknowledge this limitation; extending the evaluation to larger graphs is part of our ongoing work. We provide the results below for graphs with 20 variables, where PERI and I-PERI use PC at the client level. Due to time constraints, these results are based on 5 runs. In the next version of the paper, we will provide results over more runs and larger graphs, and include cases where GES is used at the client level.
>
> | Method               | SHD     | F1      |
> |----------------------|--------------------|-------------------|
> | PERI   | 23.20 ± 5.07       | 0.629 ± 0.081     |
> |I-PERI      | **21.40 ± 2.70**   | **0.637 ± 0.046** |
> | NOTEARS-ADMM         | 29.46 ± 6.30       | 0.490 ± 0.090     |
> | FedDAG               | 30.71 ± 5.65       | 0.225 ± 0.143     |
> | FedCDH               | 65.01 ± 15.10      | 0.286 ± 0.062     |
>
> We can observe that I-PERI is generally the best method even for larger graphs.
>
> Q: “The method strongly relies on the accuracy of local CPDAG estimation…”
>
> A: We thank the Reviewer for highlighting this point. We have explicitly reported this limitation in the final section of the paper. Errors in client level causal discovery can propagate to the federated stage and affect the final $\Phi$-CPDAG. However, this limitation is general to causal discovery methods, as the quality of the data inevitably influences the recovered graph. Nonetheless, our experiments suggest that I-PERI is robust to reasonable levels of estimation error.
>
> [6] Sachs et al. “Causal protein-signaling networks derived from multiparameter single-cell data.” Science, 2005.

---

> > ### Author Rebuttal · Reviewer_tLLZ · 2026-04-04
> >
> > All of my issues have been resolved, and I have decided to keep my score.

---

> > > ### Author Response · Authors · 2026-04-06
> > >
> > > We thank the Reviewer for the valuable questions and remain available to provide any additional clarification needed.

---

### Official Review · Reviewer_W3JC · 2026-03-13

**Soundness:** 3
**Presentation:** 3
**Significance:** 2
**Originality:** 2
**Overall Recommendation:** 4
**Confidence:** 3

**Summary:**

This paper considers the problem of federated causal discovery under unknown client-level interventions, relaxing the assumption that all clients share an identical causal model. The authors develop an approach that leverages structural differences induced by interventions across clients to recover a tigher equivalence class, called the $\phi$-Markov equivalence class. Theoretical guarantees on the convergence and privacy-preserving properties of the approach are given. Empirical studies on synthetic data are provided.

**Compliance With Llm Reviewing Policy:**

Affirmed.

**Final Justification:**

My main concerns have been addressed by the rebuttal and I have updated my score.

**Key Questions For Authors:**

Main points:
- After reading the existing works cited, it seems that Li et al. (2024) also considered similar federated causal discovery setting with unknown interventions, and provide a way to recover the causal graph. What is the improvement over it?
- It would be helpful to provide references or concrete examples where the considered structural intervention (removing only a subset of the incoming edges of a node) is relevant in practice. Otherwise, considering this intervention type may unnecessarily complicate the problem formulation without a clear motivating application.
- It would be helpful to provide examples to illustrate the differences between the proposed equivalence class and I-MEC (interventional equivalence class).

Minor points:
- There is a growing literature on federated causal discovery motivated by privacy-preserving issue. While I understand it can be challenging, it would be helpful if the authors could provide reference to concrete real applications where federated causal analysis approaches have been used in practice.
- Line 147 on Page 3: It would be helpful to provide references for "unlike most definitions of mutilated graph in the literature"

**Limitations:**

Yes

**Strengths And Weaknesses:**

Strengths:
- The problem is interesting and helps make causal discovery potentially more practical in realistic real-world scenarios.
- The theoretical result appears to be sound (although I did not carefully check all proof details), and the algorithm is reasonable, which builds upon existing regret-based federated causal discovery approach.
- The equivalence class proposed is novel to my knowledge.

Weaknesses:
- The improvement over existing work is unclear, which potentially limits the novelty of the work.
- Although the structural intervention considered is new (only a subset of the incoming edges may be removed in the interventions), it is unclear how relevant this setting is in practice as existing works mostly consider hard interventions (removing all incoming edges) or soft interventions (retaining all incoming edges) to my knowledge.
- The method only works for linear Gaussian case. Only linear Gaussian case is considered in the experiment.

---

> ### Author Rebuttal · Authors · 2026-03-31
>
> We thank the Reviewer for taking the time to provide us with valuable feedback. Any additional information in the response will be included in future versions of the paper. Following a point-by-point reply to the concerns raised by the Reviewer:
>
> **Q**: "The improvement...", "After reading...".
>
> **A**: Following, we highlight the main differences from  Li et al., 2024 (FedCDH) and the contribution of this work compared to other federated causal discovery (FCD) methods:
>
> *  In our work, we explicitly model unknown interventions at the client-level, which allows us to characterize a tighter equivalence class. In contrast, FedCDH does not explicitly model the interventions at the client-level, which limits their ability to characterize the equivalence classes.
> * In FedCDH, the type of interventions admissible is limited to soft interventions that do not alter the structure of the causal graph; this is evident by the formalization of the SCM in Section 2 and by Lemma 16 of their paper. The graph considered in their work is an augmented graph with observed domain index variables, which could represent a proxy of an intervention. In our framework, interventions are not explicitly observed.
> * We explicitly test the method with interventional data, showing we can recover a more refined graph. FedCDH only tests its method with observational data, as shown in the experimental section of their paper.
> * Unlike FedCDH, our method provides formal guarantees on differential privacy.
> * Our method is significantly faster than other methods, as shown in Section 4 of our paper.
>
> **Q**: "Although ...", It would...examples...", "Line..."
>
> **A**: We thank the Reviewer for raising this point. We are not the first to consider structural interventions that remove only a subset of the incoming edges of a node: 1) In the context of anomaly detection in IT monitoring, [2] considers interventions that remove only a subset of the incoming edges of a node; 2) In [3], authors define edge interventions, which remove only specific edges from a causal graph, describing them as a generalization of hard interventions.  We'd like to point out that our definition of structural interventions does not exclude the more classical definition of structural interventions. Moreover, as we discussed, our method can handle parametric interventions.
>
> **Q**: "It would...references..."
>
> **A**: The point raised by the Reviewer is well taken. As discussed in the paper, the $\Phi$-MEC identifies a tighter equivalence class than the classical MEC, but a looser one than the I-MEC, and the key distinction lies in how interventions are treated. The I-MEC is defined over a set of known interventions and, given sufficiently many, can recover the true DAG. Consider the first graph in the third row of Figure 4. Relying solely on observational data, we could recover only the skeleton. Now suppose we intervene on node $B$: the edge $A \rightarrow B$ is removed in the intervened graph, revealing a v-structure detectable via independence tests. Since the intervention explicitly removes an edge into $B$, and is known, we can infer that $A \rightarrow B$, recovering the full DAG. This is not possible with unknown interventions, as we cannot determine whether the intervention affects $A$ or $B$. The $\Phi$-MEC occupies an intermediate position. If we consider the federated setting in Figure 1,  the v-structure in the mutilated graph of the second client can still be identified and used to orient edges, even without knowledge of the intervention target.
>
> **Q**: “The method...”
>
> **A**: Although the experimental results are conducted on linear data, this does not limit the framework to linear settings. The method relies on the type of procedure used at the client level, which can be adapted to non-linear data, for instance (e.g., using PC with KCI). We will clarify this point and provide additional results in other settings, e.g., non-linear. For further results on real-world data, please refer to our response to Reviewer tLLZ.
>
> **Q**: "There is..."
>
> **A**: As the Reviewer noted, finding practical applications of FCD is indeed a challenge. Following two examples: 1) In [4] authors attempt to uncover clinically meaningful causal relationships in multicenter endometrial cancer datasets, producing accurate and clinically meaningful results; 2) [5] proposes a method for FCD with known interventions. The method is applied to hybrid data, real and synthetic.
>
> [2] Assaad et al. "Root cause identification for collective anomalies in time series given an acyclic summary causal graph with loops." AISTATS, 2023.
>
> [3] Shpitser et al. "Causal inference with a graphical hierarchy of interventions." Annals of Statistics, 2016.
>
> [4] Zanga et al. "Federated causal discovery with missing data in a multicentric study on endometrial cancer." JBI, 2025.
>
> [5] Abyaneh et al. "Federated Causal Discovery From Interventions.", 2022.

---

> > ### Author Rebuttal · Reviewer_W3JC · 2026-04-03
> >
> > Thanks for the response. My concerns have been addressed, and I have adjusted my score.

---

> > > ### Author Response · Authors · 2026-04-06
> > >
> > > We are grateful to the Reviewer for the detailed and constructive comments provided, and we remain available to address any additional questions or concerns that may arise.

---

### Official Review · Reviewer_odaz · 2026-03-23

**Soundness:** 2
**Presentation:** 2
**Significance:** 2
**Originality:** 3
**Overall Recommendation:** 4
**Confidence:** 3

**Summary:**

Previous papers usually assume all clients share the same causal DAG but this paper consider all clients share the same observational DAG, but the client data may sample from a unknown post-interventional DAG. It proposes I-PERI to do the two-stage causal discovery for finding CPDAG and $\Theta$-MEC.

**Compliance With Llm Reviewing Policy:**

Affirmed.

**Final Justification:**

Authors' rebuttal address most of my concern. Although they only provide the darft of the proof, it is still reasonable considering the character limit.

**Key Questions For Authors:**

See weaknesses.

**Limitations:**

yes

**Strengths And Weaknesses:**

Strengths

1. The case that client itself doesn't know the intervention of its data do exist in real world. So how to handle this problem is valuable.

2. This paper not only present some theory results but also some empirical results.

Weaknesses:

1. The proof for Theorem 3.1 (line 564), $C(G) \cap \hat{G}=C(G)$ means $C(G) \subseteq \hat{G}$, doesn't mean $C(G) = \hat{G}$. Seems the proof dose not include $\hat{G} \subseteq C(G)$.

2. For Theorem 3.3 (lines 321-325), do you mean ''for all''  instead of ''there exist''? Seems it is not aligned with the proof.

3. The symbol for intervention target seems not consistent. $\Theta_k$/$\Phi_k$/$\phi_k$ ?

---

> ### Author Rebuttal · Authors · 2026-03-31
>
> We thank the Reviewer for providing insightful feedback, taking the time to revise our work, and especially for revising the proofs.
>
> **Q**: "The proof for Theorem 3.1..."
>
> **A**: We agree with the Reviewer that the proof of Theorem 3.1 can be unclear. We realize that the convergence of $\hat{G}$ from Mian [1] does not clarify the convergence of $\hat{G}$ toward $\mathcal{C}(G)$. Nevertheless, our results still hold. The intuition is that GES, which operates at the server level, will only add edges that lead to improvements in the score. Since we start from an empty graph, we will never add edges that are not in $\mathcal{C}(G)$, or in mutilated graphs derived from it. This means that $\hat{G} \subseteq \mathcal{C}(G)$; given $n^1, \ldots, n^K \rightarrow \infty$, then $\hat{G}$ converges toward $\mathcal{C}(G)$. The proof will be extended to include, more formally, the intuition above in the next iterations of the paper. We would like to thank the Reviewer again for pointing out this issue, which is crucial for the understanding of the method.
>
> **Q**: "For Theorem 3.3..."
>
> **A**: The third point of the characterization of the equivalence classes expresses the existence of a sequence of interventions in two distinct families that create the same v-structures. What we are trying to convey is that if an intervention in the first family generates a v-structure, then there must be an intervention in the second family that generates the same v-structure in the two graphs, and vice versa. We can also phrase it in the following way: "for all interventions in the first family, there exists an intervention in the second family that generates the same v-structure in the two graphs, respectively, and for all interventions in the second family, there exists an intervention in the first family that generates the same v-structure (in the graphs)." This is equivalent to the current phrasing. We will clarify this in the paper.
>
> **Q**: "The symbol for intervention target..."
>
> **A**: We apologize for the inconsistency in the notation for intervention targets. There was an issue during the renaming of interventions: we initially used $\Theta$, then switched to $\Phi$, but some inconsistencies remained. In the paper, we use $\boldsymbol{\Phi}$ to denote a family of interventions, and $\Phi$ to denote a specific intervention within that family. We thank the Reviewer for pointing this out, and the notation will be fixed in the next version of the paper.
>
> [1] Mian, Osman, et al. Nothing but regrets - privacy-preserving federated causal discovery. AISTATS. PMLR, 2023.

---

> > ### Author Rebuttal · Reviewer_odaz · 2026-04-04
> >
> > Thanks for the detailed rebuttal. I adjust my score from 3 to 4.

---

> > > ### Author Response · Authors · 2026-04-06
> > >
> > > We are grateful to the Reviewer for the constructive discussion and the insightful comments provided. We welcome any further questions or points they would like us to clarify.

---

### Decision · Program_Chairs · 2026-04-30

**Decision:**

Accept (regular)

**Comment:**

This paper investigates federated causal discovery under unknown client‑level interventions and proposes I‑PERI, a regret‑based two-stage algorithm, building on the PERI algorithm in the literature. In the first stage, the server aggregates client regret signals to recover a global CPDAG. In the second stage, the method leverages structural differences induced by interventions across clients to orient additional edges, yielding a refined representation. The paper includes theoretical guarantees on convergence and privacy properties, and presents experimental results on synthetic datasets showing improved SHD and F1 scores compared to several baselines.

The reviewers identify several strengths of the paper. The two‑stage design is conceptually clear and interpretable. Theoretical analysis is provided under standard assumptions, and the regret‑sensitivity analysis supports the incorporation of differential privacy. The privacy‑preserving design—sharing regret values rather than raw data or local graphs—is well aligned with federated learning principles. Empirically, the method demonstrates consistent improvements over baseline methods on the reported synthetic benchmarks.

At the same time, there are notable limitations. All experiments are conducted on synthetic data, with no real‑world case studies, limiting evidence of practical applicability. The empirical setting involves relatively small graphs and a limited number of clients, leaving scalability to larger or more realistic settings insufficiently explored. In addition, the method depends critically on the accuracy of local CPDAG estimation; potential error propagation from imperfect client‑level discovery to the federated stage is not thoroughly analyzed.

These concerns, however, were addressed to a good extent in the authors' rebuttal, and all reviewers eventually converged to a verdict of weak accept.

My own reading concurs with this positive recommendation: this paper presents a well‑motivated extension of federated causal discovery, with promising theoretical and empirical results, though broader validation and deeper analysis of robustness and scalability would strengthen the contribution and clarify its practical impact.